# Low Expression of RGS2 Promotes Poor Prognosis in High-Grade Serous Ovarian Cancer

**DOI:** 10.3390/cancers14194620

**Published:** 2022-09-23

**Authors:** Jana Ihlow, Nanna Monjé, Inga Hoffmann, Philip Bischoff, Bruno Valentin Sinn, Wolfgang Daniel Schmitt, Catarina Alisa Kunze, Sylvia Darb-Esfahani, Hagen Kulbe, Elena Ioana Braicu, Jalid Sehouli, Carsten Denkert, David Horst, Eliane Tabea Taube

**Affiliations:** 1Institute of Pathology, Charité-Universitätsmedizin Berlin, Corporate Member of Freie Universität Berlin and Humboldt-Universität zu Berlin, Charitéplatz 1, 10117 Berlin, Germany; 2Berlin Institute of Health at Charité-Universitätsmedizin Berlin, Charitéplatz 1, 10117 Berlin, Germany; 3German Cancer Consortium (DKTK), Partner Site Berlin, and German Cancer Research Center (DKFZ), 69120 Heidelberg, Germany; 4Institute of Pathology, Berlin-Spandau, Stadtrandstraße 555, 13589 Berlin, Germany; 5Department of Obstetrics and Gynecology with Center of Oncological Surgery, European Competence Center for Ovarian Cancer, Charité-Universitätsmedizin Berlin, Corporate Member of Freie Universität Berlin and Humboldt-Universität zu Berlin, Campus Virchow-Clinic, Augustenburger Platz 1, 13353 Berlin, Germany; 6Tumorbank Ovarian Cancer Network, Berlin Institute of Health, Charité-Universitätsmedizin Berlin, Corporate Member of Freie Universität Berlin and Humboldt-Universität zu Berlin, Campus Virchow-Clinic, Augustenburger Platz 1, 13353 Berlin, Germany; 7Institute of Pathology, Philipps-University Marburg, Baldingerstraße, 35043 Marburg, Germany

**Keywords:** RGS2, G-protein signaling, high-grade serous ovarian cancer

## Abstract

**Simple Summary:**

Recent advances in molecular medicine have indicated G-protein coupled receptors (GPCRs) as possible therapeutic targets in ovarian cancer. The cellular effects of GPCRs are determined by regulator of G protein signaling (RGS) proteins. Especially RGS2 has currently moved into focus of cancer therapy. Therefore, we retrospectively analyzed RGS2 and its association with the prognosis of high-grade serous ovarian cancer (HGSOC). Here, we provide in situ and in silico analyses regarding the expression patterns and prognostic value of RGS2. In silico we found that RGS2 is barely detectable in tumor cells on the mRNA level in bulk and single-cell data. Applying immunohistochemistry in 519 HGSOC patients, we detected moderate to strong protein expression of RGS2 in situ in approximately half of the cohort, suggesting regulation by post translational modification. Furthermore, low protein expression of RGS2 was associated with an inferior overall- and progression-free survival. These results warrant further research of its role and related new therapeutic implications in HGSOC.

**Abstract:**

RGS2 regulates G-protein signaling by accelerating hydrolysis of GTP and has been identified as a potentially druggable target in carcinomas. Since the prognosis of patients with high-grade serous ovarian carcinoma (HGSOC) remains utterly poor, new therapeutic options are urgently needed. Previous in vitro studies have linked RGS2 suppression to chemoresistance in HGSOC, but in situ data are still missing. In this study, we characterized the expression of RGS2 and its relation to prognosis in HGSOC on the protein level by immunohistochemistry in 519 patients treated at Charité, on the mRNA level in 299 cases from TCGA and on the single-cell level in 19 cases from publicly available datasets. We found that RGS2 is barely detectable on the mRNA level in both bulk tissue (median 8.2. normalized mRNA reads) and single-cell data (median 0 normalized counts), but variably present on the protein level (median 34.5% positive tumor cells, moderate/strong expression in approximately 50% of samples). Interestingly, low expression of RGS2 had a negative impact on overall survival (*p* = 0.037) and progression-free survival (*p* = 0.058) on the protein level in lower FIGO stages and in the absence of residual tumor burden. A similar trend was detected on the mRNA level. Our results indicated a significant prognostic impact of RGS2 protein suppression in HGSOC. Due to diverging expression patterns of RGS2 on mRNA and protein levels, posttranslational modification of RGS2 is likely. Our findings warrant further research to unravel the functional role of RGS2 in HGSOC, especially in the light of new drug discovery.

## 1. Introduction

Regulator of G-protein signaling 2 (RGS2) is a member of the RGS R4 protein family and controls G-coupled receptor signaling by binding the Gα-subunit and consecutively enhancing intrinsic GTPase activity in healthy cells [1]. Furthermore, RGS2 is able to inhibit adenyl cyclase directly [2,3], to modulate muscarinic acetylcholine receptor signaling [4] and to inhibit G_q_- and G_s_-signaling, thus affecting mitogen activation protein kinase pathways (MAPK, ERK1/2) [5,6] and regulating the function of GPCRs [7]. In the past, alterations in RGS2 have been linked to cardiovascular conditions [8,9,10], regulation of insulin secretion [11], neurological disorders [12,13,14], and leukemogenesis [15,16]. Moreover, its role in solid cancer has emerged most recently, explicitly in breast carcinoma, prostate carcinoma, and ovarian carcinoma [17,18,19,20,21,22]. Since RGS-protein domains have been identified and discussed as druggable targets [23], it is relevant to analyze their biological and prognostic impact in cancer patients. This applies particularly to patients with HGSOC, whose long-term prognosis has improved with the invention of poly-ADP ribose polymerase (PARP)-inhibitors and anti-vascular endothelial growth factor (VEGF) antibodies, but is still significantly limited by chemoresistance, lack of defined therapy targets and the aggressive biology of the disease [24]. In vitro, there is growing evidence that silencing and consecutive suppression of RGS2 leads to chemoresistance in ovarian cancer cells by modulating tumor cell growth [21,22]. In situ, expression, function, and prognostic impact of RGS2 remain unknown. Since GPCRs are highly expressed in ovarian cancer [25,26,27], we explored the expression and functional role of RGS2 in HGSOC, analyzed its associations with clinicopathological characteristics, and determined the prognostic impact of RGS2 expression both in a large independent cohort of more than 500 patients on the protein level by immunohistochemistry and in silico on the mRNA level.

## 2. Patients and Methods

### 2.1. Clinical Cohort

A total of 526 patients aged ≥18 years were analyzed in this retrospective study. After diagnosis of HGSOC, patients underwent cytoreductive surgery with or without previous neoadjuvant chemotherapy at the Department of Gynecology, Charité-Universitätsmedizin Berlin, Germany between 1 January 1991 and 31 December 2019. Histology type was confirmed according to WHO criteria 2014 [28] by experienced board-approved gynecological pathologists (E.T.T., S.D.E., W.D.S., B.V.S, D.H). Apart from age < 18 years there were no further exclusion criteria, especially none that were related to the clinical presentation of the tumor. Data on overall survival (OS) were available for 519 patients. Data on progression-free survival (PFS) were available for 348 (67%) of these patients. The study was performed in accordance with the Declaration of Helsinki and with local ethical guidelines (ethic committee approval number EA1/051/18) and is supported by the TRANSCAN-2 project (grant no.: 2014-121). Clinical data were obtained from the Tumor Bank Ovarian Cancer Network (www.toc-network.de) or the Charité Comprehensive Cancer Center (https://cccc.charite.de) (accessed on 1 June 2022).

### 2.2. Tissue-Microarrays and Immunohistochemistry

Tissue microarrays (TMAs) with two tissue cores of each tumor were prepared from formalin-fixed and paraffin-embedded HGSOC tissues. For the analysis, only primary ovarian tumor tissue was used. The RGS2 antibody staining (Abcam ab36561, dilution 1:1000) was established on normal tissue using smooth muscle and colon tissue as positive controls, and liver tissue as negative control, based on the manufacturer’s instructions. HGSOC TMAs were stained immunohistochemically using a DISCOVERY XT autostainer (Ventana Medical Systems, Inc., Tucson, AZ, USA). Briefly, 5 µm TMA sections were deparaffinized, rehydrated, and subjected to heat-induced epitope retrieval followed by endogenous blocking with H_2_O_2_. Subsequently, the slides were incubated for 60 min with the RGS2 antibody (dilution 1:1000). A horseradish peroxidase (HRP)-conjugated secondary antibody was then applied for 30 min. This was followed by chromogen 3,3′-diaminobenzidine-tetrahydrochloride (DAB) application for 8 min and a counterstaining with hematoxylin and bluing reagent for 12 min.

### 2.3. Digital Image Analysis 

For digital image analysis, immunohistochemically stained TMA-slides were digitized with a Panoramic Slide Scanner (3D Histech, Budapest, Hungary), and evaluated using the open-source software platform QuPath (Version 0.2.3, available at https://github.com/qupath/qupath/releases, accessed on 05 April 2021) [29]. An automated TMA dearrayer was applied to all cores in order to identify tumor areas (TMA grid manually adjusted). After cell detection, cells were annotated as tumor cells and non-tumor cells and a two-way random trees classifier was trained for automated classification. For dichotomization, lymphocytes, macrophages, and fibroblasts were classified as non-tumor cells. An intensity threshold was set to further classify cells as negative or positive based on the mean cytoplasmic DAB density. Quality control was performed manually to exclude artefacts. Digital image analysis yielded data on the median percentage of cells with a positive staining result per total amount of tumor cells or non-tumor cells for each two cores. 

### 2.4. TCGA HGSOC Data Set Gene Expression Analysis 

Gene expression data for mRNA (RNAseq V2) and survival data were available for 299 out of 585 HGSOC patients and downloaded from cBioportal (https://www.cbioportal.org/, accessed on 5 May 2022). A ranked gene list was created by calculating Spearman’s correlations of RGS2 mRNA expression and the mRNA expression of 18,870 genes within the TCGA PanCancer Atlas dataset. Then, RGS2 and genes with the highest or lowest co-expression were visualized in a heatmap using the OncoPrint tool on cBioportal [30,31]. Furthermore, heatmaps were created considering the co-expression of maker genes for epithelial-mesenchymal transition, marker genes for methylation and genes for G-protein mediated signaling. Since only 15 genes showed a strong correlation with RGS2, a gene set enrichment analysis could not be performed.

### 2.5. Single-Cell Gene Expression Analysis in Three Publicly Available HGSOC Datasets

Single-cell analysis of RGS2 mRNA expression was performed in three different publicly available datasets [32,33,34] using the open-source software “R” (version 4.1.1, available at https://cran.r-project.org/bin/windows/base/old/4.1.1/, accessed on 10 May 2022) and the package “Seurat” (version 4.1.0, available at https://cran.r-project.org/web/packages/Seurat/index.html, accessed on 10 May 2022) [35]. Single-cell count matrices and metadata were downloaded for each dataset, and filtered for cells containing 500-6000 genes, 1000-60,000 reads, and <20% mitochondrial reads. Read counts were normalized using the scTranform function. Cells were clustered by constructing shared nearest neighbor (SNN) graphs based on the top 10 principal components at a resolution of 0.2. Uniform manifold approximation and projection (UMAP) was used for visualization. Main cell types were identified by scoring canonical cell type markers across clusters. In each dataset separately, tumor cells with RGS2 mRNA levels > or ≤ median RGS2 expression in all tumor cells were assigned as cells with high-expression of RGS2 (H-RGS2) and cells with low expression of RGS2 (L-RGS2), respectively. Differentially expressed genes in H-RGS2 and L-RGS2 tumor cells were computed using the FindAllMarkers function with the following parameters: only positive markers, fraction of expressing cells inside the cluster ≥ 0.15, difference between fraction of expressing cells inside and outside the cluster ≥ 0.15. For functional analysis, cell cycle phases were scored as implemented in “Seurat v4”. The code used for data analysis is available at https://github.com/bischofp/HGSOC_RGS2, accessed on 29 June 2022.

### 2.6. Statistical Analysis 

Patients of our cohort were grouped in H-RGS2 and L-RGS2 based on their RGS2-protein expression in tumor cells. Patients of the TCGA cohort were grouped based on their RGS2 mRNA expression in tumor cells. Cut-offs for survival analysis were defined using the digital cut-off finder of the University of Heidelberg [36]. Thresholds were identified as 5.04% positive tumor cells for protein analysis and 9.66 normalized mRNA reads for mRNA analysis. Statistical analysis was performed using IBM SPSS Statistics, Version 23 (IBM 2015, Armonk, NY, USA). Patients’ characteristics were calculated using the Mann–Whitney U test and chi square test followed by Bonferroni adjustment in multiple subgroups. OS and PFS were analyzed using the Kaplan–Meier method. To specify median follow-up, the reverse Kaplan–Meier method was applied [37]. A logrank test followed by a univariate Cox proportional hazards model was used to determine independent survival factors. To define a hazard ratio (HR), the variables were transformed into categorical dichotomous data. Factors with a significant impact on OS and PFS were analyzed in a stepwise multivariate Cox proportional hazards model. A *p*-value of <0.05 was considered statistically significant. 

## 3. Results

### 3.1. Staining Pattern of RGS2 in HGSOC

We examined the expression and distribution of RGS2 in primary HGSOC of a total of 519 patients. In the entire cohort, median RGS2 expression was 34.5% (IQR 6.4%–68.4%) in all tumor cells. Low RGS2 expression was revealed in the majority of tumor cells (Figure 1). When expressed, RGS2 was intensively and homogenously stained within tumor cells, particularly close to cell membranes. RGS2 staining intensity was approximately equal within the tumor center and the tumor edge. Opposed to areas with more solid or dissociated growth patterns, strongest and most frequent RGS2 expression was observed in papillary tumor areas, respectively (*p* < 0.001, Figure 1A,B). Of all papillary tumor areas, 62% stained strongly positive (621/1007) and 38% were negative for RGS2 (368/1007). In contrast, only 22% of all solid tumor areas were positive for RGS2 (104/455), whereas 88% were negative or had barely detectable RGS2 staining (351/455). This difference was also highlighted in samples that contained regions with a transition from papillary areas to solid or disseminated areas (Figure 1A–C). Based on a threshold of 5.04% positive tumor cells, HGSOC were then categorized as having low (L-RGS) or high (H-RGS) RGS2 expression.

### 3.2. Clinical Characteristics 

Median follow-up was 89.1 months (95% CI 77.1-101.1 months) in the entire cohort. Of all patients, the majority (88%) were diagnosed with tumor stages ≥ pT3. More advanced FIGO stages, lymphatic invasion and venous invasion accumulated in the L-RGS2 group (Table 1). Age (*p* = 0.926) and residual tumor burden (*p* = 0.699) did not differ between the H-RGS2 and the L-RGS2 group. After the initial diagnosis, all patients had received ovariectomy, radical hysterectomy, and tumor debulking. With regard to preceding neoadjuvant chemotherapy, data were available in 35% of all patients only (*n* = 184/519). Of these patients, 2 received neoadjuvant chemotherapy and 182 were untreated.

### 3.3. Low RGS2 Protein Expression Is Partially Associated with a Poor Long-Term Survival in HGSOC 

On the protein level, lower expression of RGS2 was associated with an unfavorable overall survival (*p* = 0.037) and showed a trend towards decreased progression-free survival in the univariate analysis (*p* = 0.058). Five-year OS and PFS were 40% and 14% within the H-RGS2 group as compared with 32% and 10% within the L-RGS2 group, respectively (Figure 2C,E). Both risk of death and progression were elevated by 30% in the L-RGS2 subgroup (HR 1.3). A similar trend was observed on an mRNA level within the TCGA cohort (Figure 2D,F). However, after including other relevant risk factors, such as residual tumor burden, FIGO stage, and age, this effect translated into a clear but non-significant trend towards an inferior progression-free survival (*p* = 0.193), whereas no differences in OS were observed (*p* = 0.440, Table 2).

### 3.4. RGS2 mRNA Expression in Ovarian Cancer Cells Is Weak on the Single-Cell Level

To evaluate mRNA expression in HGSOC tumor cells precisely, we analyzed RGS2 mRNA expression in silico on the single-cell level in three publicly available HGSOC data sets [32,33,34] that included data of therapy-naïve patients, patients with neoadjuvant chemotherapy and patients with metastatic disease (Figure 3A). In these datasets, a total of 19 patients were included. Nearly all tumor cells showed either weak or no RGS2 mRNA expression at all, while strong RGS2 mRNA expression was more common in stromal and immune cells, especially in macrophages (Figure 3B–D). Median RGS2 mRNA expression was 0 normalized mRNA reads per tumor cell (IQR 0.00–00) in all patients. This applied to both primary and metastatic tumor cells. In the few tumor cells with proof of RGS2-mRNA expression, RGS2 did not translate substantially into proliferative activity as it is shown by cell cycle diagrams in Figure 3C.

### 3.5. RGS2 mRNA-Suppression Is Linked to Tumor Cell Plasticity on a Single-Cell Level

Regarding the functional role of RGS2 in HGSOC, gene expression analysis was conducted in all three single-cell datasets (Figure 3E). Gene expression levels and co-expressed genes were highly variable between all datasets. Certain subtypes of L-RGS2 HGSOC held co-expression of genes coding for elements of tumor-cell integrity and tumor cell plasticity such as KLK8 (*p* = 1.13 × 10^−5^), KLK6 (*p* = 8.07 × 10^−3^) or SLPI (*p* = 2.94 × 10^−8^), CLDN7 (*p* = 5.16 × 10^−5^), SMIM22 (*p* = 1.82 ×10^−8^) and genes associated with the innate immune system pathway such as VAMP8 (*p* = 3.59 × 10^−9^), IFI27 (*p* = 2.43 × 10^−13^), and LGALS3 (*p* = 8.94 × 10^−8^). 

### 3.6. TCGA Gene Expression Analysis Reveals an Association between RGS2 Expression and Protein Synthesis and Co-Dependence on Methylation

In the TCGA cohort (*n* = 299), genes showing the strongest positive and negative co-expression with RGS2 were identified and co-expression of other functional relevant genes was explored. Median RGS2 mRNA expression was only 8.19 normalized mRNA reads (IQR 7.30–9.11) in bulk tissue. As expected, moderate co-expression of genes associated with G-Protein signaling, such as RGS1 (*p* = 1.44 × 10^−29^), GTP-binding protein GEM (*p* = 1.55 × 10^−26^), and cAMP-specific 3′,5′-cyclic phosphodiesterase 4B (PDE4B, *p* = 3.79 × 10^−22^) was observed (Table 3, Figure 4A). In general, RGS2 showed only mild correlations with other co-expressed genes with a maximum Spearman’s correlation coefficient of 0.6. Therefore, a detailed gene set enrichment analysis could not be performed. As a result of the staining patterns detected in immunohistochemistry, we examined the correlation between RGS2 expression and expression of marker genes for epithelial-mesenchymal transition (EMT, Figure 4B, Table 3). Four different expression patterns emerged: (I) RGS2 barely detectable with downregulation of classical EMT hallmark genes (SNAI1, SNAI2, ZEB1, ZEB2, VIM, FN1, and TWIST1), downregulation of CDH1 and upregulation of alternative genes for EMT (ERBB2, ERBB3, and DDR1), (II) H-RGS with downregulation of CDH1 and upregulation of EMT hallmark genes, (III) L-RGS with downregulation of CDH1 and downregulation of EMT hallmark genes, (IV) RGS2 barely detectable, downregulation of CDH1, and variable regulation of both classical and alternative EMT hallmark genes.

Due to the discrepancies between RGS2 mRNA expression and immunohistochemical RGS2 expression, further pathways of interest were evaluated. These included downstream signaling of RGS2, protein synthesis, and methylation. Interestingly, an upregulation of methylation associated marker genes was present in 60% of cases with low expression of RGS2, whereas this was not observed in the remaining cases (Figure 4C). Regarding protein synthesis, an upregulation of translation initiation factor EIF2B3 was visible in a subset of L-RGS2 patients (Figure 4D). Concerning downstream signaling of RGS2, an upregulation of PLCG1/2, PKA, and IP3 was found in certain subsets of the L-RGS2 group (Figure 4D).

## 4. Discussion

In this study, we provide novel insight into the expression patterns, functional role, and prognostic impact of RGS2 (a regulator of GPCRs), investigating it deliberately in a large cohort of HGSOC patients. We showed that RGS2 suppression has a negative impact on long-term survival on the protein level and found that RGS2 expression is lost in HGSOC with solid growth pattern. In addition, we found that RGS2 mRNA expression is related to tumor cell integrity and protein synthesis and differs substantially from RGS2 protein expression.

In our cohort, patients with RGS2 protein suppression in primary HGSOC had a substantially decreased OS and PFS. Risk of disease progression or death was increased by 30% in the univariate analysis. This effect was not observed in the multivariate analysis in the presence of residual tumor burden and advanced FIGO stages which also included metastasis. Although not significant, there was also a clear trend towards an inferior OS and PFS with RGS2 suppression on the mRNA level. Of note, AUC values were only moderate. Pre-existing data about the prognostic role of RGS2 in different cancers appear contradictory. In prostate cancer, a survival benefit has been reported in patients with low expression of RGS2 [20] similar to observations in pulmonary adenocarcinoma [38]. On the contrary, RGS2 repression seems to be linked to reduced OS in breast cancer [39], bladder cancer [40], and stage II/III colorectal cancer [41]. Due to its distinct isoforms, RGS2 has various modes of (inter)actions in these cancers [7,17,18,19,20,38,39,40,41,42]. It is mainly detected within the tumor tissue rather than in the peripheral blood. Although not essential for its evaluation as a tissue biomarker, these aspects might dampen the role of RGS2 for specific cancer monitoring.

RGS2 protein expression controls G-protein signaling in healthy cells by activating intrinsic GTPase activity [1]. Thus, our observation that RGS2 mRNA and protein expression are barely detectable in HGSOC tumor cells is reasonable from a biological point of view. Low expression of RGS2 has previously been described in chemo-resistant HGSOC cell lines. Therein promotor methylation has been revealed as the main cause for RGS2 suppression [22]. Indeed, data obtained from the TCGA cohort in our analysis showed that upregulation of methylation/acetylation-associated genes is present in approximately 75% of L-RGS2 HGSOC patients. In the remaining 25% of patients, alternative mechanisms leading to downregulation of RGS2 gene expression (such as hypoxic cell stress) are likely as it has been described in prostate cancer earlier [20]. Previous analyses indicate that RGS2 gene expression correlates directly with RGS2 mRNA expression in non-cancerous cells [43]. In contrast, our study shows a major discrepancy between RGS2 mRNA- and RGS2 protein expression, indicating post-translational modification and protein turnover as one possible mechanism of RGS2 protein regulation in HGSOC. Post-translational modification might either enhance a protective function of RGS2 in papillary HGSOC [44] or minimize it in HGSOC subtypes with solid growth, e.g., via protein kinase C dependent phosphorylation [45]. This warrants further investigations of RGS2 on the protein level, particularly in the light of recent drug discovery attempts that include targeting non-canonical mechanisms of RGS2 expression [23]. In this context, therapeutical RGS2 enhancement via selective inhibition of its proteasomal degradation might be of special interest [44,46,47].

Previous studies described the impact of RGS2 alterations on tumor cell proliferation and hormone receptor related tumor cell growth [17,19,20,40]. For HGSOC, our results indicate a multimodal function of RGS2 that is primarily related to tumor cell preservation rather than cell proliferation. In the single-cell data, proliferation and cell cycle markers were not elevated. Interestingly, we identified an association between low RGS2 expression and sustained tumor cell plasticity on the single-cell level (Figure 3D). In the H-RGS2 group, no clear expression pattern emerged. However, the single-cell data were of limited representativeness since they included only a very small number of patients (*n* = 19) which prevents from any conclusions with regard to statistical correlations. On bulk RNA levels, we recognized an activation of G-protein downstream signaling genes and an association between RGS2 expression and protein synthesis via upregulation of EIF2B3 in the L-RGS2 TCGA cohort. This supports the previous finding that RGS2 decreases global mRNA translation and protein synthesis, cellular stress response, and tumor growth by binding EIF2B3 and disrupting the EIF2-EIF2B GTPase cycle [14,48,49]. This might ultimately promote tumor cell survival and cause chemoresistance. Unlike protein synthesis, EMT seems to play a minor role in patients with RGS2 suppression, since we observed ambiguous expression patterns of EMT hallmark genes in both L-RGS and H-RGS patients on the bulk RNA level in the TCGA cohort. Yet, mechanisms for EMT might be enhanced on a protein level, since immunohistochemistry revealed a loss of RGS2 expression in tumor areas with solid growth in our own cohort. This assumption has been described previously in prostate cancer [20]. However, data from the TCGA cohort are less specific than single cell data and might be biased by the minor presence of other intratumoral cell types such as lymphocytes. Furthermore, there are several other mechanisms that additionally alter EMT and chemoresistance in HGSOC that were detected by high-throughput proteomic methods but not evaluated within the scope of this study, such as upregulation of Annexin A3 or MiR181A [50,51,52,53]. Additionally, clinical data on neoadjuvant or adjuvant chemotherapy (e.g., HiPEC) were only available in one quarter of all patients. This is a significant limitation of our study since both are known to modulate outcome and gene expression in HGSOC [54,55]. Therefore, future studies in larger cohorts should include these additional clinical data in the multivariate model. This might alter multivariate results and increase HR and significance levels.

In conclusion, RGS2 seems to be part of a multimodal protein-interaction that is associated with decreased long-term survival in primary HGSOC and might serve as a potential druggable target itself or in combination with GPCR-directed therapies, e.g., by modulating endosomal internalization of GPCR-directed nanoparticles [21,56]. So far, little is known about the RGS2-related protein interactions, the metabolomic effects of RGS2, and about its impact on cancer stem cell capacity in HGSOC. Future research on RGS2-related protein interactions should also take (neoadjuvant) chemotherapy, BRCA 1/2 mutational status, and homologue recombination deficiency (HRD) into account and may thereby shed more light onto the functional and clinical implications of RGS2 in HGSOC. This might ultimately lead to new personalized treatment options that include RGS2 as a druggable target in therapy resistant HGSOC, which remains hard to treat up until this day. The determination of tissue biomarkers in general is not sufficient to predict the clinical outcome or the presence of residual disease in patients with HGSOC. Until today, surgical effort, chemotherapy response, and center experience remain the main determining factors that improve patients’ survival [57].

## Figures and Tables

**Figure 1 cancers-14-04620-f001:**
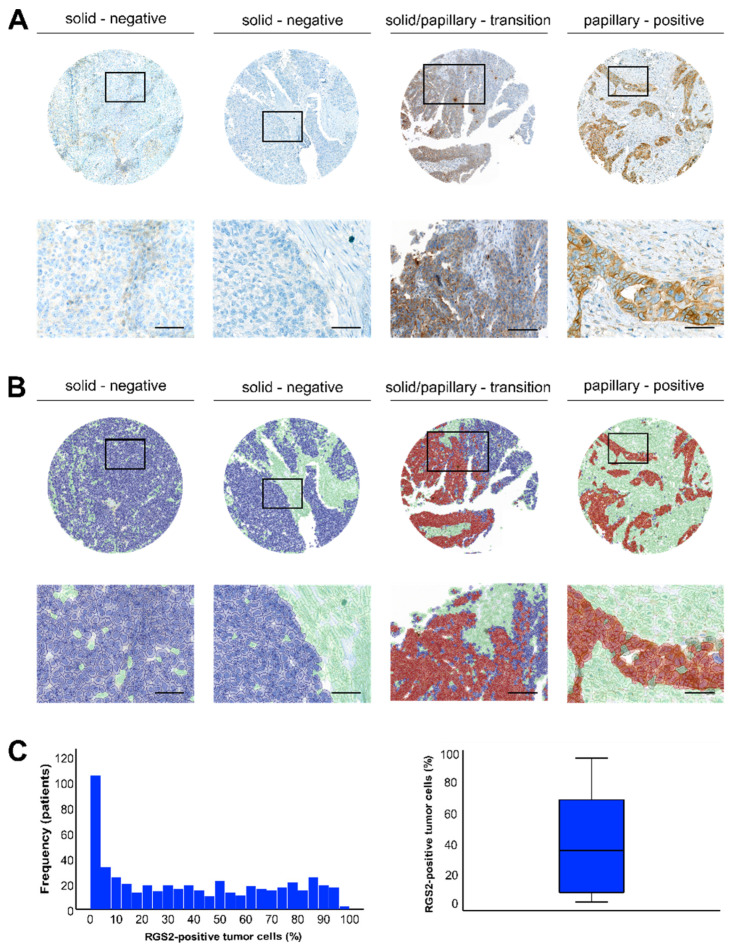
RGS2 protein expression pattern and distribution in HGSOC. (**A**) RGS2 by immunostaining in primary HGSOC with regard to growth patterns. In solid or disseminated tumor areas, RGS2 staining was negative in the majority of cases. In papillary areas, RGS2 staining was predominantly positive and distributed homogenously. In transition zones both positive and negative areas were visible. (**B**) Detected cells were color-coded according to their classification: green = non-tumor cells, blue = RGS2-negative tumor cells, red = RGS2-positive tumor cells. Panel images are magnifications of boxed areas in upper panel images. Scale bars (left → right): Picture 1 and 2: 120 µm, Picture 3: 250 µm, Picture 4: 80 µm. (**C**) Distribution of RGS2-positive tumor cells in the entire cohort. Most tumor cells were RGS2 negative or showed weak RGS2-expression; however, approximately half of the patients showed detectable RGS2-expression in ≥50% of tumor cells.

**Figure 2 cancers-14-04620-f002:**
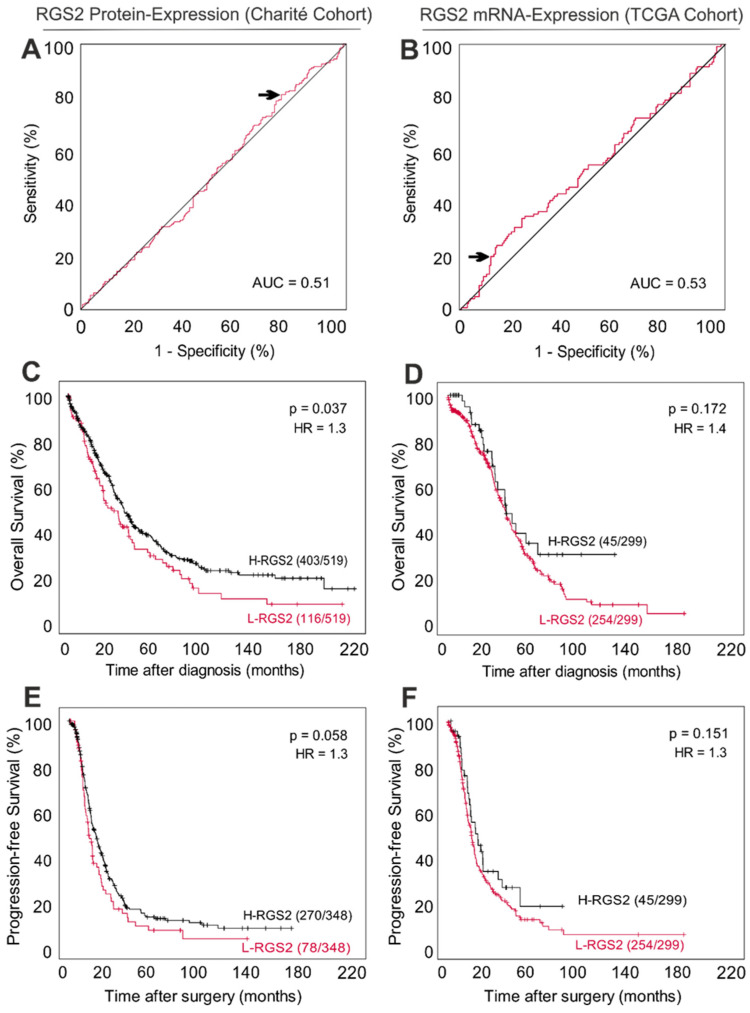
Low RGS2 expression indicates poor survival in HGSOC. (**A**) ROC curve determining best discrimination thresholds of RGS2 protein expression (percent of tumor cells) in the HGSOC Charité-cohort. (**B**) ROC curve determining best discrimination threshold of RGS2 mRNA Expression in the HGSOC TCGA cohort. The arrow indicates chosen value for binary classification. AUC, area under curve. (**C**,**E**) Overall- and progression-free survival for patients with low and high RGS2 protein expression within the Charité cohort. (**D**,**F**) Overall and progression-free survival for patients with low and high RGS2 mRNA-expression within the TCGA cohort. *p*-value indicates log rank test result, L-RGS2/H-RGS2: low/high expression of RGS2. HR: hazard ratio.

**Figure 3 cancers-14-04620-f003:**
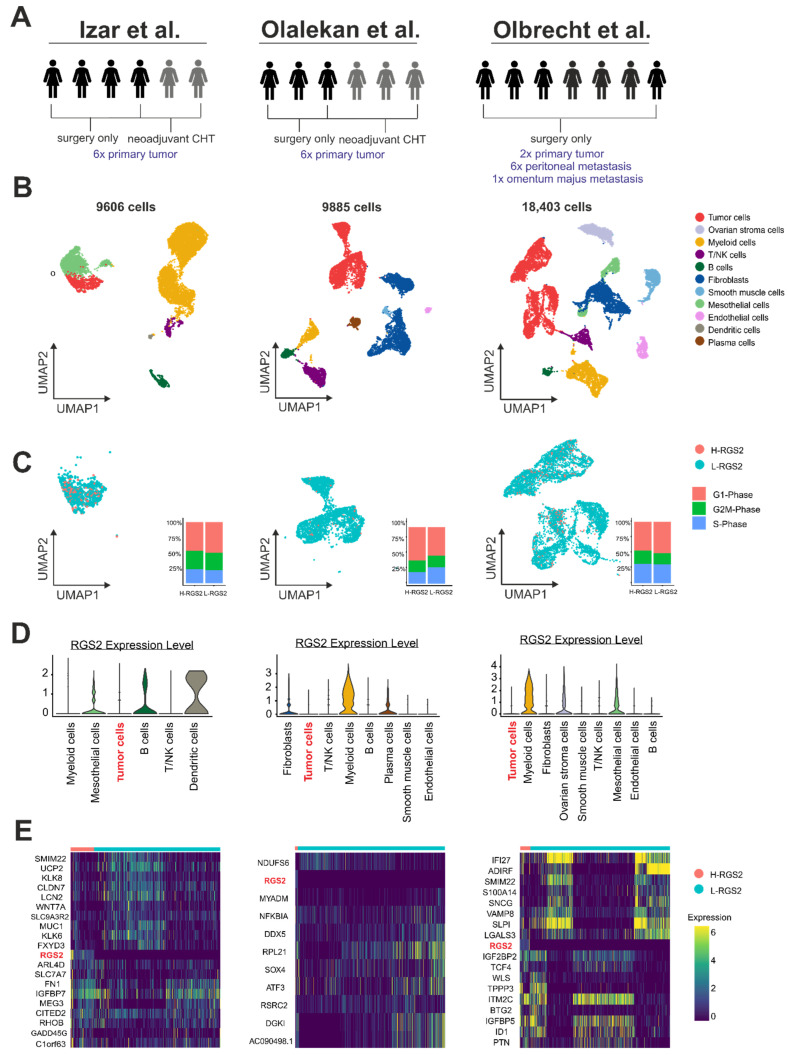
Single cell analysis with regard to RGS2 mRNA expression in three pre-existing datasets. (**A**) Composition of each HGSOC study cohort and tissue sampling sites, each manikin represents one patient. CHT: chemotherapy [32,33,34] (**B**) Number and composition of cell types within each HGSOC study cohort. Each dot represents one cell. (**C**) RGS2 mRNA expression within HGSOC tumor cells and associated cell cycle phases. Each dot represents one tumor cell. Most of the tumor cells show low RGS2 expression and cell cycle phases do not differ substantially between cells with low or high RGS2 expression (L-/H-RGS2). (**D**) mRNA expression within each cell type that was present per dataset. As compared with surrounding non-malignant cells, RGS2 mRNA expression is markedly low within the tumor cell compartment. (**E**) Differential mRNA expression of genes showing strongest or weakest correlation with RGS2 in the tumor cell compartment within each dataset. Each row represents one cell.

**Figure 4 cancers-14-04620-f004:**
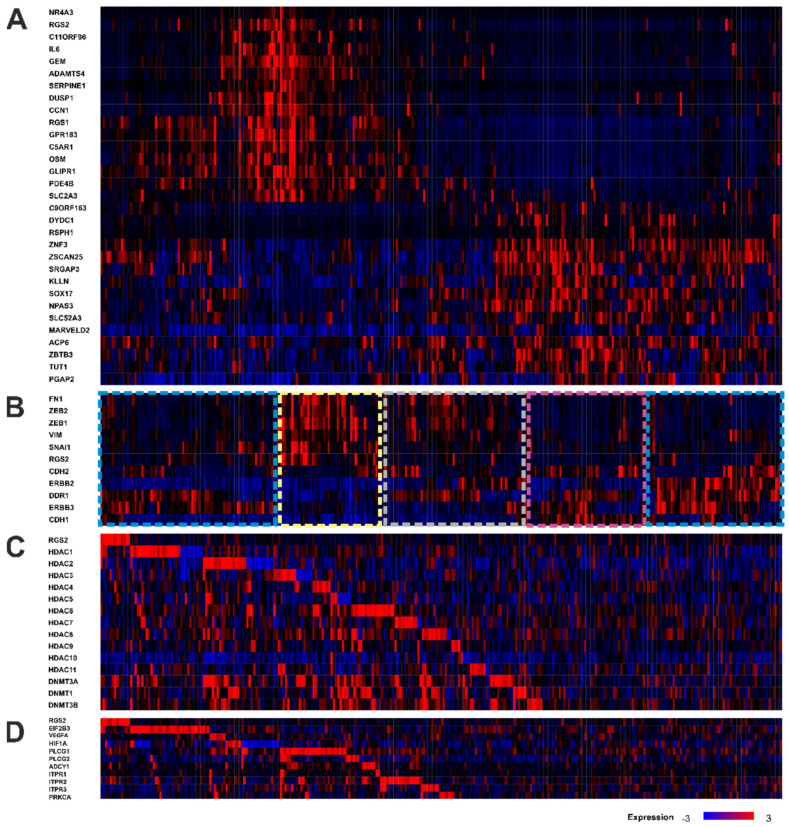
Bulk mRNA expression data from the TCGA HGSOC cohort (*n* = 299) for functional relevant genes in the context of RGS2 expression obtained from cBioportal. (**A**) Differential expression of the TOP15 genes with strongest and weakest co-expression with RGS2. RGS2 shows moderate co-expression of genes associated with signaling or pro-inflammatory states and negative correlation with genes coding for zinc finger domains. (**B**) RGS2 and co-expression of maker genes for epithelial-mesenchymal transition (EMT) reveals four different clusters: I. RGS2 barely detectable, downregulation of classical EMT hallmark genes and upregulation of alternative genes for EMT (blue boxes on the left and right), II. H-RGS and upregulation of EMT hallmark genes (yellow box), III. L-RGS and downregulation of EMT hallmark genes (pink box), IV. RGS2 barely detectable and variable regulation of both classical and alternative EMT hallmark genes (grey box). (**C**) RGS2 and co-expression marker genes for methylation reveals increased methylation activity more than 50% of patients with low RGS2 expression. (**D**) RGS2 and co-expression of genes associated with RGS2 downstream signaling, hypoxic cell stress and alternative signaling mechanisms demonstrates post-translational modification (EIF2B3), hypoxic cell stress (HIFα) and compensatory upregulation of downstream targets in nearly half of the patients with low RGS2 expression and underlines the positive feedback loop between RGS2 suppression and phospholipase C (PLCG1 and 2) mediated phosphorylation of RGS2.

**Table 1 cancers-14-04620-t001:** Clinical characteristics and univariate survival in the HGSOC cohort with regard to RGS2 protein expression.

Characteristics	Entire Cohort	L-RGS2	H-RGS2	*p*-Value
*n* (%)	519	116 (22)	403 (78)	
Age (years), median (IQR)	61.5 (53–69)	62 (55–68)	61 (53–70)	0.926
RGS2-positive tumor cells, median, % (IQR)	34.5 (6.4–68.4)	1.1 (0.3–2.4)	49.2 (24.2–75.4)	**<0.001**
Primary tumor stage (pT)				0.092
- pT1, *n* (%)	31 (6)	5 (4)	26 (6)	
- pT2, *n* (%)	32 (6)	3 (3)	29 (7)	
- pT3, *n* (%)	456 (88)	108 (93)	348 (87)	
Lymph node stage (pN)				0.140
- pN0, *n* (%)	127 (24)	23 (20)	74 (18)	
- pN1, *n* (%)	290 (56)	74 (64)	216 (54)	
- pNX, *n* (%)	102 (20)	19 (16)	290 (72)	
Distant metastasis (pM)				0.104
- pM0, *n* (%)	211 (41)	43 (37)	168 (42)	
- pM1, *n* (%)	105 (20)	30 (26)	75 (18)	
- pMX, *n* (%)	203 (39)	43 (37)	160 (40)	
FIGO stage				0.185
- FIGO I, *n* (%)	24 (5)	5 (4)	19 (5)	
- FIGO II, *n* (%)	22 (4)	2 (2)	20 (5)	
- FIGO III, *n* (%)	373 (72)	80 (69)	293 (73)	
- FIGO IV, *n* (%)	100 (19)	29 (25)	71 (17)	
Residual tumor burden, *n* (%)				0.699
- present, *n* (%)	133 (26)	32 (27.5)	101 (25)	
- not present, *n* (%)	236 (45)	52 (45)	184 (46)	
- n.A., *n* (%)	150 (29)	32 (27.5)	118 (29)	
OS, median (95% CI)	41.2 (36.1–46.3)	30.6 (21.0–40.1)	43.0 (37.3–48.7)	**0.037**
PFS (*n*= 348 pts.), median (95% CI)	19.3 (16.9–21.7)	15.9 (12.3–19.6)	19.3 (16.9–21.7)	0.058

Abbreviations: regulator of G-protein signaling (RGS2), low RGS2 expression (L-RGS2), high RGS2 expression (H-RGS2), number of patients (*n*), interquartile range (IQR), Fédération Internationale de Gynécologie et d’Obstretique (FIGO), not available (n.A.), overall survival (OS), progression free survival (PFS), confidence interval (CI), patients (pts). Bold font indicates statistical significance.

**Table 2 cancers-14-04620-t002:** Multivariate Cox regression model for OS and PFS in HGSOC patients with regard to immunohistochemical RGS2 protein expression in tumor cells and other factors with univariate significance.

	OS (*n* = 519)				PFS (*n* = 348)	
Variable	HR	95% CI	*p*	HR	95% CI	*p*
Age > 60	1.16	0.96–1.51	0.233	1.02	0.78–1.33	0.912
FIGO > II	2.08	1.02–4.24	**0.045**	1.90	0.94–3.90	0.076
Residual tumor	2.10	1.61–2.72	**<0.001**	1.64	1.22–2.20	**0.001**
L-RGS2	1.24	0.84–1.51	0.440	1.24	0.90–1.71	0.193

Abbreviations: overall survival (OS), progression-free survival (PFS), number of patients (*n*), hazard ratio (HR), confidence interval (CI), Fédération Internationale de Gynécologie et d’Obstretique (FIGO), low expression of RGS2 (L-RGS2). Bold font indicates statistical significance.

**Table 3 cancers-14-04620-t003:** Correlation between RGS2 mRNA-Expression and mRNA expression of associated genes in the TCGA cohort (*n* = 299). A: TOP 15 genes that are positively correlated with RGS2. B: TOP 15 genes that are negatively correlated with RGS2. C: Co-expression of hallmark-genes for epithelial, mesenchymal translation. D: Hallmark genes for methylation. E: Genes that are associated with downstream signaling of RGS2 and hypoxia. Bold font indicates statistical significance.

Gene	Localization	Spearman’s Correlation	*p*-Value
**A**			
C5AR1	19q13.32	0.591	**1.24 × 10^−29^**
RGS1	1q31.2	0.590	**1.44 × 10^−29^**
NR4A3	9q22	0.587	**3.64 × 10^−29^**
GPR183	13q32.3	0.584	**8.27 × 10^−29^**
DUSP1	5q35.1	0.576	**6.98 × 10^−28^**
OSM	22q12.2	0.574	**1.24 × 10^−27^**
IL6	7p15.3	0.564	**1.28 × 10^−26^**
GEM	8q22.1	0.563	**1.55 × 10^−26^**
ADAMTS4	1q23.3	0.550	**4.03 × 10^−25^**
C11ORF96	11p11.2	0.547	**7.71 × 10^−25^**
SERPINE1	7q22.1	0.547	**7.82 × 10^−25^**
CCN1	1p22.3	0.538	**7.01 × 10^−24^**
GLIPR1	12q21.2	0.521	**3.10 × 10^−22^**
PDE4B	1p31.3	0.520	**3.79 × 10^−22^**
SLC2A3	12p13.31	0.514	**1.38 × 10^−21^**
**B**			
ZNF3	7q22.1	−0.329	**5.15 × 10^−9^**
C9ORF163	9q34.3	−0.315	**2.40 × 10^−8^**
ZBTB3	11q12.3	−0.309	**4.83 × 10^−8^**
ZSCAN25	7q22.1	−0.301	**1.11 × 10^−7^**
MARVELD2	5q13.2	−0.300	**1.15 × 10^−7^**
NPAS3	14q13.1	−0.292	**2.63 × 10^−7^**
RSPH1	21q22.3	−0.284	**5.88 × 10^−7^**
SLC52A3	20p13	−0.283	**6.28 × 10^−7^**
DYDC1	10q23.1	−0.280	**8.02 × 10^−7^**
KLLN	10q23	−0.278	**9.90 × 10^−7^**
PGAP2	11p15.4	−0.276	**1.12 × 10^−3^**
SOX17	8q11.23	−0.274	**1.48 × 10^−3^**
SRGAP3	3p25.3	−0.273	**1.57 × 10^−3^**
TUT1	11q12.3	−0.269	**2.18 × 10^−3^**
ACP6	1q21.2	−0.269	**2.19 × 10^−3^**
**C**			
VIM	10p13	0.293	**2.47 × 10^−7^**
ZEB2	2q22.3	0.424	**1.55 × 10^−14^**
ZEB1	10p11.22	0.409	**1.59 × 10^−13^**
FN1	2q35	0.402	**4.54 × 10^−13^**
SNAI2	8q11.21	0.343	**1.05 × 10^−9^**
TWIST1	7p21.1	0.389	**2.65 × 10^−12^**
SNAI1	20q13.13	0.427	**1.06 × 10^−14^**
CDH2	18q12.1	−0.019	0.749
ERBB2	17q12	−0.055	0.343
DDR1	6p21.33	−0.090	0.118
ERBB3	12q13.2	−0.153	**7.92 × 10^−3^**
CDH1	16q22.1	−0.104	0.072
**D**			
HDAC6	Xp11.23	−0.142	**0.014**
HDAC11	3p25.1	−0.135	**0.019**
HDAC7	12q13.11	−0.135	**0.020**
HDAC4	2q37.3	−0.111	0.054
HDAC1	1p35.2-p35.1	−0.083	0.154
HDAC3	5q31.3	−0.078	0.179
HDAC5	17q21.31	0.077	0.186
HDAC10	22q13.33	0.054	0.355
HDAC2	6q21	0.051	0.379
HDAC9	7p21.1	−0.015	0.796
HDAC8	Xq13.1	−0.007	0.900
DNMT1	19p13.2	−0.010	0.085
DNMT3A	2p23.3	0.048	0.410
DNMT3B	20q11.21	−0.028	0.625
**E**			
EIF2B3	1p34.1	0.137	**0.018**
VEGFA	6p21.1	0.109	0.061
HIF1A	14q23.2	0.039	0.503
PLCG2	16q24.1	0.081	0.164
PLCG1	20q12	9.46 × 10^−3^	0.870
ITPR1	3p26.1	0.112	0.053
ITPR3	6p21.31	−0.085	0.142
ITPR2	12p11.23	0.049	0.398
ADCY1	7p12.3	−0.028	0.634
PRKCA	17q24.2	0.116	0.148

Bold font indicates statistical significance.

## Data Availability

Data are available from the corresponding author upon reasonable request.

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
