# Peer review of "Low Expression of RGS2 Promotes Poor Prognosis in High-Grade Serous Ovarian Cancer"

_cancers, 2022, doi:10.3390/cancers14194620_

Round 1

Reviewer 1 Report

Well written manuscript. Interesting data. HR <2 not really meaningful.  Perhaps sample size can be increased in another study and additional clinical data be used in the model that increases HR.

Author Response

We added this aspect to the discussion (page 13, lines 294-295 and page 15, lines 351-353).

Reviewer 2 Report

The goal of this study was to evaluate regulator of G protein signaling 2 (RGS2) expression for its prognostic association in high grade serous ovarian cancer (HGSOC). Immunohistochemistry staining was performed on tissues from patients treated at Charité, and both single cell and bulk RNA analyses were conducted from public datasets. RGS2 is expressed in a variety of cancers, and it is yet to be determined if this is a potential therapeutic target in oncology. While the novelty of the study is appreciated, there are several critical flaws that reduce the impact of this study.

The staining of “solid/papillary – transition” tissues appears non-specific and lacks cell surface localization as depicted in the “papillary – positive” specimen. Given the staining is utilized to distinguish low and high RGS2 expression, the lack of clear and consistent staining precludes downstream comparisons and analyses.

The study lacks adequate statistical significance. In figure 2, AUC values are not strong. RGS2 staining is only modestly correlative with overall survival in the Charité cohort, and is not significant in the TCGA cohort. Progression-free survival is also not significant in either cohort. Outside of RGS2 expression, which is the metric used to determine the grouping, there are no significant clinical characteristics associated with RGS2 in Table 1.

It is unclear how RGS2 mRNA is undetectable/extremely low in tumor samples from single cell datasets, while intratumoral immune cells, fibroblasts, and mesothelial cells show higher mRNA expression. This should be clarified or discussed, as it does not follow the protein staining data in previous figures.

Additionally, previous cited studies suggest that RGS2 suppression modulates tumor cell growth, however proliferation/cell cycle markers are not altered in the single cell mRNA data. This should be addressed and discussed.

The bulk RNAseq data does not discriminate immune infiltrates or stromal fibroblast/mesothelial cells, and thus it is impossible to determine the source of RGS2 mRNA in this dataset. Based on the single-cell data, the highest expression of RGS2 is not in the tumor cell fraction. Therefore, the correlation data in Figure 4 and the data in Table 3 can not be confidently attributed to tumor cell expression. Additionally, there is a lack of analysis of overlap in RGS2-associated genes from the single-cell data in Figure 3E and the bulk RNA data in Figure 4.

Collectively, the data presented here is correlative with a lack of strong statistical significance to make substantial claims. The data lacks cohesiveness, and does not support the indicated hypothesis that RGS2 is a promising drug target for HGSOC.

Author Response

The goal of this study was to evaluate regulator of G protein signaling 2 (RGS2) expression for its prognostic association in high grade serous ovarian cancer (HGSOC). Immunohistochemistry staining was performed on tissues from patients treated at Charité, and both single cell and bulk RNA analyses were conducted from public datasets. RGS2 is expressed in a variety of cancers, and it is yet to be determined if this is a potential therapeutic target in oncology. While the novelty of the study is appreciated, there are several critical flaws that reduce the impact of this study.

1. The staining of “solid/papillary – transition” tissues appears non-specific and lacks cell surface localization as depicted in the “papillary – positive” specimen. Given the staining is utilized to distinguish low and high RGS2 expression, the lack of clear and consistent staining precludes downstream comparisons and analyses.

Answer: The staining was reviewed independently by 5 board-approved pathologists of our institute (E.T.T., S.D.E., W.D.S., B.V.S and D.H). We have chosen another area from the same sample, with more specific staining in the papillary area (Figure 1A and B, third row each). In areas with weaker staining, positivity and negativity of the staining was revealed through the digital classification that is highly reliable.

2. The study lacks adequate statistical significance. In figure 2, AUC values are not strong. RGS2 staining is only modestly correlative with overall survival in the Charité cohort and is not significant in the TCGA cohort. Progression-free survival is also not significant in either cohort. Outside of RGS2 expression, which is the metric used to determine the grouping, there are no significant clinical characteristics associated with RGS2 in Table 1.

Answer: We agree that AUC levels are not optimal in our cohort and added that aspect to the discussion (page 13, line 296). We also added the circumstance, that the adverse effect of RGS2 suppression does not translate significantly on the mRNA level. However, there was a clear and visible trend towards an inferior OS in these data as well. We have added this aspect to the discussion (page 13, lines 294-296). As you pointed out below, there is some limitation to the exactness of bulk RNA seq data due to mixed cell types within one sample. This might also impact the sharpness of the survival analysis as compared to protein analysis since protein data result from specific manual and digital annotation of tumor cells. With regard to baseline characteristics our data demonstrate that the Charité Cohort is very homogenous and thus RGS2 expression is fortunately not biased by any other important baseline characteristic that could interfere the analysis.

3. It is unclear how RGS2 mRNA is undetectable/extremely low in tumor samples from single cell datasets, while intratumoral immune cells, fibroblasts, and mesothelial cells show higher mRNA expression. This should be clarified or discussed, as it does not follow the protein staining data in previous figures.

Answer: We have stated in the discussion that the single cell data are restricted to a very small number only (n=19) and might therefore be less representative (page 14, lines 330-333). This was the initial reason for adding a TCGA cohort analysis. However, it is the power of single cell sequencing to uncover exactly these effects. As discussed on page 14, lines 315 ff., we assume that DNA-modification, hypoxic cell stress, post-translational modification, and protein turnover may play major mechanistical roles in the differences between RNA and protein expression of RGS2. This is supported by earlier findings [1-5].

4. Additionally, previous cited studies suggest that RGS2 suppression modulates tumor cell growth, however proliferation/cell cycle markers are not altered in the single cell mRNA data. This should be addressed and discussed.

Answer: We agree and have added this aspect to the discussion (page 14, lines 325 ff). Previous studies have been performed in bulk tissue only but have stated that the role of RGS2 seems somewhat contradictory, especially with regard to RGS2-related effects in tumor cell growth. By using the single cell technique in further cohorts, we will be able to unravel new aspects on this topic and gather more information in the future.

5. The bulk RNAseq data does not discriminate immune infiltrates or stromal fibroblast/mesothelial cells, and thus it is impossible to determine the source of RGS2 mRNA in this dataset. Based on the single-cell data, the highest expression of RGS2 is not in the tumor cell fraction. Therefore, the correlation data in Figure 4 and the data in Table 3 can not be confidently attributed to tumor cell expression. Additionally, there is a lack of analysis of overlap in RGS2-associated genes from the single-cell data in Figure 3E and the bulk RNA data in Figure 4.

Answer: We agree that associated genes differ in single cell analysis and bulk RNA datasets. This is puzzling since genes detected in bulk RNA seq should be refined in single cell data with regard to tumor cells and tumor microenvironment. Usually, tumor acquisition for bulk RNA data is conducted after macroscopic (or microscopic) selection of pure tumor areas. Thus, data obtained from bulk RNA seq should represent mostly tumor cells. On the other hand, single cell data might lack representativeness due to small and inconsistent cohorts (n=19). Further scRNAseq-studies addressing the allocation of genes in HGSOC samples would be very interesting. We have added this aspect to the discussion (page 14, lines 330-333, and page 15, lines 344-346).

6. Collectively, the data presented here is correlative with a lack of strong statistical significance to make substantial claims. The data lacks cohesiveness, and does not support the indicated hypothesis that RGS2 is a promising drug target for HGSOC.

Answer: Certainly, our study has some limitations that we have added to the discussion. Nevertheless, we do believe that our data warrant further investigation of mechanisms regulating RGS2 expression in HGSOC which is still hard to target and has only limited treatment options. Whether RGS2 is a promising drug target in HGSOC should be investigated by further studies that need to include in-vitro evidence and prospective in-vivo data. We have mitigated the title a little (“RGS2 promotes poor survival in HGSOC”).

References

1. Sjögren B, Parra S, Heath LJ, Atkins KB, Xie Z-J, Neubig RR. Cardiotonic Steroids Stabilize Regulator of G Protein Signaling 2 Protein Levels. Mol Pharmacol. 2012;82(3):500.

2. Cunningham ML, Waldo GL, Hollinger S, Hepler JR, Harden TK. Protein kinase C phosphorylates RGS2 and modulates its capacity for negative regulation of Galpha 11 signaling. J Biol. Chem. 2001;276(8):5438-44.

3. Alqinyah M, Hooks SB. Regulating the regulators: Epigenetic, transcriptional, and post-translational regulation of RGS proteins. Cell Signal. 2018 Jan;42:77-87.

4. Linder A, Hagberg Thulin M, Damber J-E, Welén K. Analysis of regulator of protein signalling 2 (RGS2) expression and function during prostate cancer progression. Sci Rep. 2018;8(1):17259.

5. Cacan E. Epigenetic regulation of RGS2 (Regulator of G-protein signaling 2) in chemoresistant ovarian cancer cells. J Chemother. 2017;29(3):173-8.

Reviewer 3 Report

This is a manuscript of paramount soundness that serves the motivation to identify putative future biomarkers to tackle hard-to-treat cancers, such as HGSOC. There are several novelties acknowledged in the manuscript, amongst which I consider, for instance, the improvement of the IHC data reliability score, which makes it of added value in the field of academic research or the new evidence on the inhibition of mRNA translation into protein. The authors need to elaborate on their choice of RGS2 for expression and functional analysis in HGSOC. RGS2 is a multifaceted protein with various modes of action; this diversity, due to the distinct functional isoforms could be what makes it potentially a low cancer-specific biomarker.  I reckon the choice is because EGFRs and GPCRs are highly expressed in ovarian carcinoma. That needs mentioning in the introduction with a reference. Nevertheless, there are certain caveats about such a choice despite the evidence at the protein level and detection in most ovarian cell lines; it is not detected in the blood by immunoassay or mass spectrometry and in addition, its " metabolomic " effect, if any have not been explored. Whether it increases cancer stem cell capacity can also be an interesting direction for research.

Ultimately, and to give the manuscript a more clinical orientation for the clinical audience, an argument needs to be made that tissue-based biomarkers cannot reliably predict classical HGSOC prognosticators, such as residual disease. Surgical effort, surgical skills, and center experience remain the determining factors alongside chemotherapy response to improved patient survival (with a reference). 

Author Response

This is a manuscript of paramount soundness that serves the motivation to identify putative future biomarkers to tackle hard-to-treat cancers, such as HGSOC. There are several novelties acknowledged in the manuscript, amongst which I consider, for instance, the improvement of the IHC data reliability score, which makes it of added value in the field of academic research or the new evidence on the inhibition of mRNA translation into protein.

1. The authors need to elaborate on their choice of RGS2 for expression and functional analysis in HGSOC. RGS2 is a multifaceted protein with various modes of action; this diversity, due to the distinct functional isoforms could be what makes it potentially a low cancer-specific biomarker.  I reckon the choice is because EGFRs and GPCRs are highly expressed in ovarian carcinoma. That needs mentioning in the introduction with a reference.

Answer: We have added these aspects to the introduction (page 4, lines 93,105-106) and to the discussion (page 15, lines 354-357).

2. Nevertheless, there are certain caveats about such a choice despite the evidence at the protein level and detection in most ovarian cell lines; it is not detected in the blood by immunoassay or mass spectrometry and in addition, its " metabolomic " effect, if any have not been explored. Whether it increases cancer stem cell capacity can also be an interesting direction for research.

Answer: We agree and have added these aspects to the discussion (page 13, lines 302-304 page 15, lines 357-359).

3. Ultimately, and to give the manuscript a more clinical orientation for the clinical audience, an argument needs to be made that tissue-based biomarkers cannot reliably predict classical HGSOC prognosticators, such as residual disease. Surgical effort, surgical skills, and center experience remain the determining factors alongside chemotherapy response to improved patient survival (with a reference). 

Answer: We agree and have added these aspects to the discussion (page 15-16, lines 363-367, reference No.57).

Reviewer 4 Report

I read with great interest this Manuscript, which falls within the aim of the Journal.
Honestly, the topic is interesting enough to attract the readers’ attention. The methodology is accurate, and the data analysis supports conclusions. Nevertheless, authors should clarify some points and improve the discussion by citing relevant and novel critical articles about the topic.

- OVERALL COMMENTS:

- A native English speaker should further revise Manuscript to improve clarity and readability. Punctuation also needs to be corrected. (Numerous sentences end without a period).

- I want to inform You that I make a plagiarism check routinely, and I can confirm that Yours is an original writing

- METHODS:

- Although it is a retrospective analysis, Inclusion/exclusion criteria should be better clarified by extending their description. In particular, I suggest adding a section dedicated to the clinical characteristics of the patients. It is well reported in Table 1, but I suggest specifying that were no restrictions for the clinical presentation of the tumor. It should also be added a consideration about the kind of tissue analyzed (Ovarian tissue? Carcinomatosis? Both?)

-RESULTS:

The result section should not contain a term that implies a personal consideration (Line 246: Interestingly); (Line 274: Interestingly)

Line 243 should be moved to discussion

- DISCUSSION:
- The authors have not adequately highlighted the strengths and limitations of their study. I suggest better specifying these points
- What are the actual clinical implications of this study? it is essential to report the results obtained by the authors in the context of clinical practice and to adequately highlight what contribution this study adds to the literature already existing on the topic and to future study perspectives
- In my opinion, one of the significant biases of the study is the lack of data regarding the distinction between patients who did or did not receive neoadjuvant chemotherapy, which is known to alter the chemoresistant status of ovarian cancer. I suggest you devote a few lines of the discussion to this concept.

-In addition, it should also be explained how in the case of IDS, the possible use of HIPEC may impact OS and tumor molecular expression (pls see PMID: 32715605; PMID: 32931024. The absence of data regarding the distribution of these approaches between the two patient groups may represent a Bias and should also be reported in the results if possible.

-Likewise, it should also be emphasized that it is not the object of the study to weigh the weight that the various mRNA modifications present in the ETM may exert. However, to improve the quality of the writing, it would be necessary to report this as a possible bias and contextualize it with the existing literature (pls see PMID: 21435174; PMID: 33228245; PMID: 17269733; PMID: 27249598)

Author Response

I read with great interest this manuscript, which falls within the aim of the Journal. Honestly, the topic is interesting enough to attract the readers’ attention. The methodology is accurate, and the data analysis supports conclusions. Nevertheless, authors should clarify some points and improve the discussion by citing relevant and novel critical articles about the topic.

1. A native English speaker should further revise Manuscript to improve clarity and readability. Punctuation also needs to be corrected. (Numerous sentences end without a period). I want to inform You that I make a plagiarism check routinely, and I can confirm that Yours is an original writing.

Answer: As recommended, we have had the manuscript spell-checked by a native English speaker who is working at our institute (Ryan Blathnáid). Mrs Ryan’s help has been mentioned in the acknowledgement.

2. Although it is a retrospective analysis, Inclusion/exclusion criteria should be better clarified by extending their description. In particular, I suggest adding a section dedicated to the clinical characteristics of the patients. It is well reported in Table 1, but I suggest specifying that were no restrictions for the clinical presentation of the tumor. It should also be added a consideration about the kind of tissue analyzed (Ovarian tissue? Carcinomatosis? Both?)

Answer: We have added the fact that this is a retrospective analysis both in the simple summary and in the methods section (page 2, lines 39 and page 5, line 117). Exclusion criteria (age <18) have been clarified in the methods section (page 5, lines 122-124). Furthermore, the type of tissue used for this analysis was mentioned in the methods section (primary ovarian carcinoma tissue, page 5, lines 132-133). As suggested, we have added a short paragraph with focus on clinical characteristics to the results section (page 9, lines 216-224).

3. The result section should not contain a term that implies a personal consideration (Line 246: Interestingly); (Line 274: Interestingly)

Answer: The term “interestingly” as removed from the results section.

4. Line 243 should be moved to discussion.

Answer: Line 243 was moved to the discussion (now on page 14, lines 330-331).

5. DISCUSSION: The authors have not adequately highlighted the strengths and limitations of their study. I suggest better specifying these points:
- What are the actual clinical implications of this study?

- It is essential to report the results obtained by the authors in the context of clinical practice and to adequately highlight what contribution this study adds to the literature already existing on the topic and to future study perspectives

Answer: The clinical implication of this study was to examine RGS2 as a potential biomarker for HGSOC and to discuss RGS2 or RGS2-related pathways as a potential druggable targets in HGSOC. RGS2 was chosen on purpose, because it regulates G-protein signaling of GPCR-subtypes, that are frequently (over)expressed in HGSOC. Due to this aspect, GPCRs themselves have been discussed as potential therapeutical targets for HGSOC, in particular for intraperitoneal application. RGS2-modulation itself might even enhance intraperitoneal internalization of nanoparticles targeting affiliated GPCRs. We have outlined this in the introduction (page 4, line 105-106) and discussion (page 15, lines 355 ff.). We have also added the fact, that this is the first study investigating RGS2 in a larger cohort of HGSOC patients deliberately and emphasized the clinical context (page 13, lines 286-287 and page 15, lines 355 ff.).

6. In my opinion, one of the significant biases of the study is the lack of data regarding the distinction between patients who did or did not receive neoadjuvant chemotherapy, which is known to alter the chemoresistant status of ovarian cancer. I suggest you devote a few lines of the discussion to this concept.

Answer: We agree and have added this limitation to the results (page 9, lines 222-224) and discussion sections (page 15, lines 349 ff.). Neoadjuvant chemotherapy in HGSOC is not recommended within German treatment standards [1-5]. Therefore, it is assumable that the majority of patients did not receive it. In our cohort, definitive information on neoadjuvant chemotherapy is available in 184 patients only. Of these patients, only 2 received neoadjuvant chemotherapy, which reflects the uncommon application of it. Therefore, we could not include this factor into the multivariate survival analysis but at least our data shouldn’t hold a strong bias.

7. In addition, it should also be explained how in the case of IDS, the possible use of HIPEC may impact OS and tumor molecular expression (pls see PMID: 32715605; PMID: 32931024. The absence of data regarding the distribution of these approaches between the two patient groups may represent a Bias and should also be reported in the results if possible.

Answer: This is a very interesting aspect. In our center HIPEC is not applied on a regular basis, therefore we were not able to generate data on this issue. It would be very interesting, to examine, how RGS2 is modulated by HIPEC therapy in HGSOC or even whether future intraperitoneal application of GPCR- and RGS2 directed therapy could potentially have a potent effect on ovarian cancer that has not metastasized beyond the peritoneal cavity. We have added this aspect to the discussion and listed the interesting references that you mentioned (page 15, lines 349-351).

8. Likewise, it should also be emphasized that it is not the object of the study to weigh the weight that the various mRNA modifications present in the ETM may exert. However, to improve the quality of the writing, it would be necessary to report this as a possible bias and contextualize it with the existing literature (pls see PMID: 21435174; PMID: 33228245; PMID: 17269733; PMID: 27249598).

Answer: We have added these aspects and the related literature to the discussion (page 15, lines 346-348).

References

1. German S3 guideline, version 5.1 for therapy and aftercare of malignant ovarian cancer, page 76 , available at https://www.leitlinienprogramm-onkologie.de/leitlinien/ovarialkarzinom/, accessed: 2.9.2022

2. Vergote, I., et al., Neoadjuvant chemotherapy or primary surgery in stage IIIC or IV ovarian cancer. N Engl J Med, 2010. 363(10): p. 943-53.

3. van der Burg, M.E., et al., The effect of debulking surgery after induction chemotherapy on the prognosis in advanced epithelial ovarian cancer. Gynecological Cancer Cooperative Group of the European Organization for Research and Treatment of Cancer. N Engl J Med, 1995. 332(10): p. 629-34

4. Rose PG, et al., A phase III randomized study of interval secondary cytoreduction in patients with advanced stage ovarian carcinoma with suboptimal residual disease: a Gynecologic Oncology Group study. ASCO, 2002.

5. Redman, C.W., et al., Intervention debulking surgery in advanced epithelial ovarian cancer. Br J Obstet Gynaecol, 1994. 101(2): p. 142-6.

Round 2

Reviewer 4 Report

Dear Authors,

Thank You for the slight corrections, which improved the quality of the article.